# Temporal Gene Expression Profiles Reflect the Dynamics of Lymphoid Differentiation

**DOI:** 10.3390/ijms23031115

**Published:** 2022-01-20

**Authors:** Smahane Chalabi, Agnes Legrand, Victoria Michaels, Marie-Ange Palomares, Robert Olaso, Anne Boland, Jean-François Deleuze, Sophie Ezine, Christophe Battail, Diana Tronik-Le Roux

**Affiliations:** 1Centre National de Recherche en Génomique Humaine, CEA, Université Paris-Saclay, 91057 Evry, France; smahane.chalabi@gmail.com (S.C.); palomares@cng.fr (M.-A.P.); olaso@cng.fr (R.O.); boland@cng.fr (A.B.); deleuze@cng.fr (J.-F.D.); christophe.battail@cea.fr (C.B.); 2Laboratoire Européen de Recherche pour la Polyarthrite Rhumatoïde—Genhotel, Université Paris-Saclay, Univ Evry, 91042 Evry, France; 3INSERM, Unité 1151, Université De Paris, Unité Mixte de Recherche 8253, 75015 Paris, France; agnes.legrand@inserm.fr (A.L.); vicky0678@gmail.com (V.M.); 4Laboratoire Biologie et Biotechnologies pour la Santé, IRIG, UMR 1292 INSERM-CEA-UGA, Univ. Grenoble Alpes, 38000 Grenoble, France; 5Atomic Energy and Alternative Energies Agency (CEA), Department of Research in Hemato-Immunology (SRHI), Saint-Louis Hospital, 75010 Paris, France; diana.le-roux@cea.fr; 6IRSL, HIPI-UMRS 976, University of Paris, 75006 Paris, France

**Keywords:** lymphopoiesis, hematopoietic differentiation, RNAseq, long non-coding RNA, transcriptome, gene networks

## Abstract

Understanding the emergence of lymphoid committed cells from multipotent progenitors (MPP) is a great challenge in hematopoiesis. To gain deeper insight into the dynamic expression changes associated with these transitions, we report the quantitative transcriptome of two MPP subsets and the common lymphoid progenitor (CLP). While the transcriptome is rather stable between MPP2 and MPP3, expression changes increase with differentiation. Among those, we found that pioneer lymphoid genes such as *Rag1*, *Mpeg1*, and *Dntt* are expressed continuously from MPP2. Others, such as CD93, are CLP specific, suggesting their potential use as new markers to improve purification of lymphoid populations. Notably, a six-transcription factor network orchestrates the lymphoid differentiation program. Additionally, we pinpointed 24 long intergenic-non-coding RNA (lincRNA) differentially expressed through commitment and further identified seven novel forms. Collectively, our approach provides a comprehensive landscape of coding and non-coding transcriptomes expressed during lymphoid commitment.

## 1. Introduction

Hematopoietic stem cell (HSC) differentiation is a continuous process in which stem cells gradually lose their self-renewal capacity to generate hierarchically organized subpopulations of heterogeneous multipotent progenitors (MPPs) with decreasing self-renewal potential, which ultimately give rise to all mature blood cells. All these populations display different signatures that are mainly defined according to the presence or absence of a limited number of surface markers. The scarcity of the known markers prevents a fully homogeneous purification.

In mouse bone marrow (BM), all HSCs and MPPs reside in the lineage negative (Lin^−^) Sca-1^+^c-Kit^hi^ (LSK) compartment (about 0.1% of all BM cells). Within this population, MPP subsets are described according to Slam antigens [1] or the expression of VCAM1 and Flt3 surface markers [2]. According to the latter classification, three subsets of MPPs can be identified. MPP1 (VCAM1^+^Flt3^−^) the most immature, can generate all blood cell lineages. MPP2 (VCAM1^+^Flt3^+^) has lost megakaryocyte/erythroid potential and MPP3 (VCAM1^−^Flt3^+^) has lost myeloid potential and gives rise to the common lymphoid progenitors (CLPs) [3]. This more restricted progenitor (Lin^−^Sca-1^low^c-Kit^low^IL7Ralpha^+^) has lost the ability to produce myeloid cells in vivo but is able to generate mainly B cells, the most abundant lymphoid subset produced in the BM but also T cells, dendritic cells (DC), Natural Killers (NK), and Innate Lymphoid Cells (ILCs) [3,4].

Microarrays studies have provided useful insights into HSC and MPP biology [5,6,7]. Similarly, single-cell RNA (scRNA) sequencing has helped advance our understanding of the fate of hematopoietic cells [8,9,10,11,12]. However, the limited sensitivity of currently available scRNA approaches, capturing only the expression of a few thousand genes per cell, does not allow quantification of the entire protein-coding and non-coding transcriptomes. Despite these experimental improvements, the trajectories of MPPs to more committed populations and particularly to lymphoid cell populations, are still unresolved [13].

Several key molecules, such as transcription factors (TFs), are reported to orchestrate complex and fine-tune regulatory networks involved in the hematopoietic differentiation. TFs instruct lineage fate decisions and control cell transcriptomes as cells proliferate, develop, and respond to their environments. They trigger the activation or silencing of transcriptional programs through their binding to specific sequences in regulatory regions, called TF binding sites (TFBSs). Noteworthy, deregulation or chromosomal translocations of TFs that act on the hematopoietic system disturb the homeostatic balance and prompt human malignancies [14] emphasizing the need to extensively identify their dynamic patterns of expression. Several reviews provide insights into the functions of some critical TFs involved in different hematopoietic lineages [15,16,17,18]. Some of these such as the C/EBP family, PU.1, E2A, IKAROS, and FOXO1 are general TFs [19]; whereas TCF7, E2A, IRF8, BCL11B are more specifically involved in the development of lymphoid cells [20]. Elucidation of the sequential emergence of these factors and their organization into gene regulatory networks would be a major step towards gaining a mechanistic understanding of cellular decision-making processes.

Another class of gene expression regulators, not extensively studied yet, include the long non-coding RNAs (lncRNA), which are RNA transcripts without protein-coding capacity that act at the transcriptional and post-transcriptional level. They have multifaceted roles in the field of physiology and the development of disease and have emerged as important constituents of regulatory networks [21,22].

In the hematopoietic system, some lncRNAs control progenitor differentiation [23,24,25,26,27]. Other lncRNAs are involved in erythroid maturation and erythro-megakaryocyte development [28,29], or associated with myeloid committed progenitors such as *H19*, *Meg3*, and *Malat1* [30]. However, the full analysis of lncRNAs involved in the commitment of lymphoid progenitors is still to be performed. Hence, a current challenge is to perform the full count of lncRNAs and to translate their temporal expression modifications into molecular functions.

In this context, we carried out deep RNA-sequencing of the whole transcriptome of three different populations at specific stages of the lymphoid differentiation trajectory to order lymphoid gene expression through commitment from multipotent progenitors (MPP2 and MPP3) to the next committed subset, CLP. Our findings revealed specific and dynamic expression profiles for the different cell populations as the result of the action of TFs regulatory networks. Some of the newly identified markers may help to refine the purification of progenitors. Finally, we expanded the current repertoire of annotated and yet unidentified lncRNAs expressed in lymphoid progenitors. Altogether, the data provide novel gene expression information that may help a better understanding of the programs involved in the commitment of hematopoietic progenitor cells to the lymphoid lineage.

## 2. Results

### 2.1. Whole-Transcriptome Profiling of Hematopoietic Progenitors

Transcriptome analysis was performed to gain deeper insight into the temporal expression changes associated with the commitment of multipotent progenitors (MPPs) to the lymphoid lineage. To this end, MPP2, MPP3, and CLP populations were isolated by fluorescence-activated cell sorting (FACS) from the BM of B6 mice based on the surface expression of VCAM1, FLT3, and IL7Ralpha within the LSK population (Figure 1A,B). We aimed to analyze the total population of CLP to be able to have access to the different potentialities of this progenitor.

On average 100 K cells were analyzed for each population in three independent biological experiments (Appendix A). We generated sequence libraries from these low RNA inputs using the SMARTer Stranded RNAseq and sequenced at high depth of coverage to generate ~2 × 120 M reads per biological replicate.

Quality controls performed on raw and aligned reads were in line with what is expected for total RNA sequencing (Appendix A). In particular, the distribution of aligned reads on the mouse genome indicated that ~25% of alignments were localized in gene intronic regions. We observed a higher proportion of duplicated reads (62% on average), compared to classical RNA sequencing data. This is due to the low quantity of total RNA engaged in sequence library preparation. However, our high depth of coverage achieved per biological replicate allowed us to compensate for this read duplication rate. For further analysis, we only eliminated one replicate (MPP3 replicate 3) because of its very low read alignment rate (31% of aligned reads on mouse genome with only 5% of unique reads).

Transcriptome quantification (as described in the gene quantification panel, Appendix A), identified on average 10,742, 11,019, and 10,480 expressed genes (FPKM > 0.1 in at least one biological replicate before ComBat batch effect correction [31]) for MPP2, MPP3, and CLP populations, respectively (Appendix A). Therefore, nearly the same proportion of genes is expressed in each progenitor. Principal component analysis (PCA) performed on this set of corrected genes expressed in each biological replicates revealed that the transcriptomes of the three hematopoietic progenitors were clearly separated (Figure 1C). Furthermore, PCA showed the nearby clustering of cell replicates. We also assessed reproducibility of transcriptomes generated from biological replicates by unsupervised clustering of expressed genes before and after batch effect correction (Appendix A). We found that the removal of batch effects, due to the technical variability associated with the different batches of mice, significantly improved the clustering.

Classification of expressed genes into RNA categories indicated that on average 85% and 2% of expressed genes are protein-coding and long non-coding genes (lncRNA), respectively (Figure 1D). Moreover, distribution of expressed genes into RNA categories is highly reproducible within biological replicates and among progenitors, revealing the high suitability of the RNAseq data to be interrogated to obtain a comprehensive landscape of protein-coding and lncRNA genes involved in lymphoid commitment.

### 2.2. Identification of Novel Temporal Genes Expression Profiles Associated with Lymphoid Commitment

To identify gene expression changes that occur during the transition MPP2-MPP3-CLP, we first performed pairwise comparisons of protein-coding gene abundances. We detected in total 1890 differentially expressed genes (DEGs) (Appendix A). The distribution of DEGs among MPP2 vs. MPP3, MPP3 vs. CLP, and MPP2 vs. CLP comparisons was 83, 1111, and 1771 DEGs, respectively (|Log_2_ Fold Change| > 0.5, FDR = 0.05). This reveals the stability of transcriptional programs expressed by MPP2 and MPP3, and the strong shift in expression at the MPP3-CLP transition (Appendix A).

A detailed analysis of the few DEGs identified at the MPP2-MPP3 transition showed that they were in majority over-expressed in MPP3 (Figure 2A top panel). As for the DEGs identified between the MPP2/MPP3 and CLP stages, they were distributed in a balanced manner between over- and under-expressed genes (Figure 2A middle and bottom panels, respectively). The comparison of the DEGs revealed that 91% of the MPP2 vs. MPP3 DEGs (76 out of 83 genes) and 90% of the MPP3 vs. CLP DEGs (995 out of 1111 genes) were also identified by the MPP2 vs. CLP transition (Figure 2B).

To better understand the differentiation process, we analyzed the main DEGs associated with each stage. We first selected DEGs between MPP2 and MPP3 with |Log_2_(MPP3/MPP2)| > 1.5 (Figure 2C). Among the top 17 genes, nine were associated with the lymphoid immune system pathway (*Ly86*, *Dntt*, *Dtx4*, *Rag1*, *Rag2*, *Arpp21*, *Zap70*, *Clec2i*, *Bfsp2*) suggesting that these nine genes represent the earliest lymphoid genes already expressed in MPP.

Regarding the top DEGs between MPP3 and CLP (Figure 2D, |Log_2_ (CLP/MPP3)| > 3), we found that the cytoskeleton and adhesion genes were the most highly modulated. These categories include upregulated genes such as *Myl4*, *Epha2*, *Fgr*, *Amica1* and downregulated genes such as *Col4a2*, *Col16a1*, *Scarf1*. This probably reflects the circulation/move of CLP towards the lymphoid niche, as recently suggested [32]. In addition, CLP highly express several genes of the immune system, such as the B cell specific genes *Blnk*, *Siglech*, *Ebf1*, *Rag1*, *IgII1*, *Vpreb1*, *Vpreb2*, *Vpreb3*, *Gfra2*, and *Mpeg1*. Moreover, *Gimap4*, a specific target gene of *Pu1* within T-cell-progenitors is also detected [33].

To identify markers that emerge specifically in CLP, we look for genes very weakly expressed in MPP2 and over-expressed in CLP (mean (Log_2_ MPP2) < 10^−2^ FPKM and |Log_2_ (CLP/MPP2)| > 1.5). Among the 23 genes detected, 19 were B and T cell specific genes (Figure 2E), demonstrating the presence of both programs in CLP. Applying an additional filter that removes undetectable genes in the MPP3 population reveals a short list of 10 genes (*Igll1*, *6330403A02Rik*, *Vpreb1*, *Cecr2*, *Vpreb3*, *Ebf1*, *Gfra1*, *Bst1*, *Klrk1*, *Akap12*) that are specifically expressed in CLP.

To test further the robustness of the RNAseq results, we selected a set of six genes (CD7, HES1, BLNK, EPHA2, VPREB1, and F2RL3) to perform the RT-PCR validation. To this end, all three progenitors were sorted at a concentration of 10 cells/well. For each population, eight of these wells were independently amplified by RT-PCR. The results of the PCR amplification are shown in Figure 3A. The *y*-axis corresponds to the percent of positive wells (0.1 correspond to 10% whereas 1 correspond to 100% of positive wells). Therefore, the temporal expression profiles obtained for these genes are in accordance with their RNAseq quantification (Figure 3A, Appendix A). Moreover, specific transcripts encoding well-known markers were also preferentially expressed in the expected population, showing further the robustness of the RNAseq results (Appendix A). In particular, *Flt3*, *Notch1*, *Ccr9*, *Cxcr4*, *Il7ralpha*, *Ets1*, and *Gata3* were over-expressed in CLP cells compared to MPPs, and *c-Kit* was over-expressed in MPPs compared to CLP, consistent with its downregulation in lymphoid cells [34].

Altogether, the data demonstrate that the expression of lymphoid genes is initiated in MPP2 and remains active along the lymphoid differentiation ending up in a fully lymphoid CLP population expressing a new set of specific genes.

### 2.3. Exploratory Analysis of MPP-to-CLP-Lineage Specific Surface Markers

To further identify specific cell surface markers (SM) that ultimately might improve the purification of progenitor cells, we performed hierarchical clustering on the temporal gene expression profiles of the 677 SMs differentially expressed in the three progenitors (MPP2-MPP3-CLP). Genes were separated into two groups depending on whether their expression values increase or decrease from MPP2 to CLP. Both categories included comparable numbers of deregulated genes (341 upregulated versus 336 downregulated) (Appendix A).

Among the SMs over-expressed in CLP and almost undetectable in MPP2, we found B cell specific such as *Vpreb1*, *Bfsp2*, *Gfra2;* T/NK cell specific such as *Klrk1*, *Zap70*, *Icos*, *Cd28*, *Tcf7*; and pan-lymphoid such as *Slamf6* (Figure 2E).

Notably, the expression of CD34, *c-Kit* and *Csf3r*, the granulocyte colony stimulating factor receptor are downregulated in MPP2 suggesting that one important aspect of lymphoid engagement is the downregulation of receptors that are important for myeloid lineages [35].

We then restricted our analysis to the 87 SMs corresponding to known clusters of differentiation (CD) (https://www.genenames.org/data/genegroup/#!/group/471 (accessed on 4 March 2019) (Figure 3B). From these, FACS analysis pinpointed CD93 (also known as AA4.1) as a good candidate. Consistent with the RNAseq results, this marker was not significantly detected in the MPP populations (MPP2 = 0.9% and MPP3 = 1.6%) whereas the expression in CLP was high (about 72%) (Figure 3C top panel).

To get a better insight into the role of CD93, the CLP population was split into two subsets according to the CD93 expression (CD93^+^ and CD93^−^) and their capacity to generate B or T cells was assessed on OP9 and OP9-DL4 stroma, respectively. The sorted CD93^+^ and CD93^−^ CLP were seeded and cultured for 21 days. After this period, FACS analysis showed that both subsets do generate the full B lineage (assessed with anti-B220) whereas only CD93^−^ cells generate T cells that develop into CD4/CD8 populations in OP9-DL4 stroma (Figure 3C middle and bottom panels). Altogether, the results show that the CLP is composed of two different populations based on the expression of CD93 and experimentally confirm the specific expression of CD93 in CLP compared to MPP2 and MPP3, showing that it can be used as an additional marker to improve the purification of these populations.

### 2.4. Identification of Biological Pathways through Lymphoid Commitment

We continued to explore the DEGs by searching for biological pathways activated or repressed during lymphoid commitment in order to identify the key pathways that trigger the commitment to CLP.

Biological pathways activated or repressed in CLPs, identified by ontological enrichment analysis performed using the PantherDB tool (version 14.1 Released 2019-03-12), are represented in Figure 4 and Appendix A. Top activated categories include genes involved in the establishment of the immune system, such as B and T cell activation. In addition, we found the activation of EGF receptor, FGF, VEGF, PDGF, and p38 MAPK signaling pathways.

In contrast, the major biological pathways repressed in CLPs were involved in metabolism (pentose phosphate, fructose, glycolysis). Notably, as glycolysis is a key player in the maintenance of stemness through provision of energy [36], its repression is logically associated with the loss of stemness property during the differentiation of MPP cells into CLP.

Some biological pathways such as those associated with inflammation (chemokine, cytokine, and interleukin signaling pathways) or cell differentiation (integrin, endothelin, and insulin-like growth factor-1 signaling pathways) were enriched in both, up- and downregulated genes, indicating the complex regulation of these pathways during progenitor differentiation (Appendix A).Notably, comparison of our transcriptome analysis with two publicly available datasets (Cabezas-Wallscheid et al., 2014 [37]; Klimmeck et al., 2014 [30]) reveal that the most enriched pathways were associated with cytokine regulation, cell cycle regulators, and stress-associated genes (Appendix A). Among cytokines, it was interesting to highlight IL18 and *Il18r1*, the cytokine receptor associated with the strongest over-expression (Figure 5A). *Il18r1* is expressed in a variety of immune cells including thymic ETP, NK cells, macrophages, neutrophils, B cells, and fully differentiated Th1. Furthermore, it was also detected in B cell disorders [38]. Indeed, IL-18 can positively impact BM lymphopoiesis and T cell development, presumably via interaction with the c-Kit and IL-7 signaling axis. Furthermore, IL-18 can substitute for IL-7 in early thymic development processes and can synergize with IL-7 in promoting proliferation [39]. Two chemokines, *Ccr2* and *Cxcr4*, over-expressed in CLP were also highlighted (Figure 5B), reflecting trafficking and de-attachment of newly generated progenitors from the stroma to reach the circulation [40,41,42].

In addition, consistent with an active cell cycle status, the quiescence-enforcer Cdkn1c (p57, a regulator of HSC quiescence) is found mostly expressed in MPP2 compared to CLP (Figure 5C) [43]. Conversely, CLPs clearly showed robust expression levels of the cell-cycle driver *Cdk6*, *Cdk8*, *Cdk19*, *Cdk5r1* compared to MPPs (which express *Cdk18*, closely linked to actin [44]).

Concerning the mechanisms protecting cellular integrity, we found a novel set of Serpins (*Serpinb8*, *Serpinb6b*, *Serpinb9*, *Serpinf1*, *Serpina3g*) over-expressed in MPPs compared to CLP (Figure 5D).

### 2.5. Prediction of a Gene Regulatory Network Related to Lymphoid Differentiation

To identify the regulatory networks controlling lymphoid differentiation, we first analyzed the expression profiles of TFs across the MPP2-CLP transition.

We detected 28 TFs over-expressed in CLP (Log_2_(CLP/MPP2) > 1) (Appendix A), which include TF specific for the T and NK lineages such as *Gata3*, *Hes1*, *Tcf7*, *Eomes*; TFs specific for DC and B cells such as *Ebf1*, *Foxo1*, *Klf4*, and *Irf8*; and TF such as *Ets1*, *Tcf3*, *Ikzf1-Ikaros*, *Pou2f1/Oct1*, *Sp3* that act on several lymphoid lineages. This confirms the multilineage developmental potential of CLP [8].

Regarding the 25 downregulated TFs in CLP (Log_2_ (CLP/MPP2) < −1) (Appendix A), we notably identified genes specific to non-lymphoid lineages, such as those involved in erythroid and myeloid/megakaryocytes development (*Gata2*, *Hoxa10*, *Hlf*, *Nfe2*, *Nfix*) and genes important for stem cell differentiation (*Lmo2*, *Tal1*) (Appendix A).

To obtain a more dynamic view of the differentiation, we inferred networks of co-regulated genes that have common TFBS. These were searched in promoter regions of the 758 co-expressed SMs and TFs. For this motif enrichment analysis, we used the AME method from MEME suite and the collection of TFBS position frequency matrices referenced in the HOCOMOCO mouse database [45,46]. TFBS motifs for SPI1, PLAG1, ELF1, FOXO1, FOXM1, MEF2D, and STAT3 were predicted as significantly enriched in promoter regions of co-expressed genes (Appendix A). Since SPI1 and PLAG1 are downregulated TFs, we focus on their under-expressed target genes that are associated with the repression of gene expression programs. Consistent with previous literature reports, in this network we found *c-Kit* and *Ebi3*. Notably, 35 genes are regulated simultaneously by both TFs (Figure 6). The other genes included in this network are newly described. The network of upregulated genes controlled by *Elf1*, *Foxo1*, *Foxm1*, and *Mef2d* is involved in the positive regulation. In this case, such factors control the expression of multiple essential regulators involved in T and B cell biology.

In conclusion, six main TFs orchestrate the network of genes whose expression varies during commitment of MPP2 to CLP (Figure 6). The robustness of these results is supported by an extensive literature analysis showing that 61 TF–target gene associations that correspond to 12% of the total network connections identified here are already published in other models (Appendix A). Altogether, the integrated analysis of co-expression and regulatory binding site prediction allowed us to identify a broad number of co-regulated genes associated with lymphoid differentiation, which highlights the complexity and fine regulation of the process.

### 2.6. Annotated and Novel Long Non-Coding RNAs Associated with Lymphoid Progenitor Differentiation

To further characterize molecular events involved in the differentiation of the lymphoid lineage, we explored the temporal expression of lncRNAs.

We first focused on lincRNA with known genomic annotation. From the total RNA sequencing data generated, we detected 353 lincRNAs expressed in the three progenitors MPP2, MPP3, and CLP (Appendix A), from which 23 were differentially expressed between MPP2 and CLP cells (Figure 7A and Appendix A). Nine of them were also differentially expressed by both MPP2 vs. CLP and MPP3 vs. CLP comparisons, but not between MPP2 and MPP3 (Appendix A). Again, this shows a certain MPP2-MPP3 stability followed by significant expression changes during the transition of MPP3 to CLP. Their precise function in the emergence of the CLP merits further investigation. Interestingly, *Snhg7* is the only lincRNA that has a different expression profile: it expression first increases from MPP2 to MPP3 before decreasing significantly at the CLP stage.

Another set of 14 differentially expressed lincRNAs were only found in the transition from MPP2 to CLP, but not from MPP3 to CLP (Figure 7A and Appendix A), suggesting a role in other hematopoietic lineages.

To go beyond the analysis of annotated lincRNAs, we searched for novel lncRNAs. To this end, we carried out genome-guided transcript assembly using Cufflinks tool and selected assembled transcripts which did not overlap annotated mouse genes on the same strand (GRCM38 Ensembl release 75). Given that more than 96% of novel assembled transcripts were mono-exonic, we decided to focus our analysis on this category. We assessed the coding potential of each novel transcript using the CPAT tool [47] and kept putative novel long non-coding RNA greater than 200 bp in length. We then extracted raw read counts for these transcripts using HTSeq-count tool and identified differentially expressed transcripts with DESeq2 software. A short list of 45 novel lncRNAs predicted by this methodology were equally distributed between intergenic and intronic regions (Appendix A).

To obtain an estimate of the biological pertinence of these novel lncRNAs, we examined their predicted secondary structure. The calculation and visualization of the MFE (minimum free energy) structures were obtained using the RNAfold tool. The results indicated that these novel lncRNAs could be modeled with stable secondary structures (MFE < 80 kcal/mol) (Figure 7B), consistent with a potential biological function [48].

Finally, to assess the relevance of our predictions, we carried out the experimental validation of six novel lncRNAs (three in intergenic and three in intronic regions) by RT-PCR (Figure 7C, Appendix A). The results show that these novel lncRNAs are indeed expressed in these progenitors and validate, for two of them (LNC13 and LNC84), their upregulation during the differentiation towards the CLP.

## 3. Discussion

A major step towards gaining a mechanistic understanding of cellular decision-making processes is to determine what the phenotypes of cells are, at different transition stages, and to establish cell type-specific regulatory networks that control the spatiotemporal expression of lineage-specific genes.

In this study, we tracked lymphoid commitment through a comprehensive RNAseq analysis on three purified progenitor populations: MPP2-MPP3-CLP, to highlight transcriptional modifications occurring through the gradual loss of multilineage potential to reach the fully committed lymphoid progenitor.

The RNAseq data reveal that the transcriptional program is rather stable between MPP2 and MPP3 cells followed by a strong shift in expression at the MPP3-CLP transition. Notably, while lymphoid gene expression is barely detectable in HSC [49], our analysis reveals that MPP2 cells express pioneer lymphoid lineage restricted genes such as *Rag1*, *Mpeg1*, and *Dntt* which remain active along the lymphoid differentiation towards the CLP population. These results are consistent with previously published expression data showing that the abundance of these three genes is barely detectable in the HSC and MPP1 stages and then increases from MPP2 [37]. *Rag1* and *Dntt* are recombinases and transferases, respectively, involved in the B and T-cell receptor recombination [50]. *Mpeg1* is a myeloid gene whose expression is detected on zebra fish B cells [51]. However, for a complete lymphoid priming the upregulation of key lymphoid TF such as *Ebf1*, *Tcf7*, *Spib*, *Hes1*, *Eomes*, and genes associated with both B cell (*VpreB1/3*, *CD93*) and T cell receptor signaling (*Zap70*) is mandatory.

The transcriptome analysis reveals a pluri-lymphoid (T, B, DC, NK) signature of CLP, even though the main fate of this progenitor is to generate the B cell lineage, and sequentially, the other lineages [4]. Our analysis is consistent with the plasticity of such progenitors that have diverse open options immediately available. The final decision might be undeniably dependent on the microenvironment [52]. For example, CLP in the BM generate B cell progenitors, however, injected intra thymically, they produce mainly T cells [4]. Collectively, the data presented here illustrate coexistence of multiple competing lineage-specific transcriptional programs and reveal novel signaling programs not previously identified as operating during lymphoid commitment.

We also identified genes, such as CD93, that are highly expressed in CLP cells compared to MPP2 and MPP3. CD93 is a cell-surface glycoprotein and type I membrane protein that is involved in intercellular adhesion and in the clearance of apoptotic cells [53]. It is required for maintenance of antibody secretion and persistence of plasma cells in the bone marrow niche [54]. The differential expression of CD93 suggests that it can be used as additional marker to improve the purification of progenitors. Notably, we observed differences within the CLP population: the CD93^+^ and CD93^−^ cells generate the full B lineage whereas only CD93^−^ cells generate T cells that develop into CD4/CD8 populations in OP9-DL4 stroma.

In this study, bioinformatic analyses enable to identify networks of interacting TFs which in the case of multilineage differentiation may give rise to multiple distinct outcomes. In particular, we found that TFs such as *Spi1*, *Plag1*, *Elf1*, *Foxo1*, *Foxm1*, and *Mef2d* regulate sets of co-expressed related genes most of them reported for the first time. The fact that very well known transcription factors, such as *Gata3*, *Notch1*, or *c-kit* are well classified in each category adds more validity to the work. More importantly, we identified a broad number of co-regulated genes associated with lymphoid differentiation, which might help unravel new aspects of this complex process.

In addition, computational screening identified 353 lincRNAs expressed in lymphoid progenitors with known genomic annotation, of which 24 were DEGs. The level of expression of nine of those was similar in MPP2 and MPP3 cells but highly modulated during the MPP3-CLP transition (Figure 7) suggesting a role of these lincRNAs in the establishment of the CLP population. A set of 14 differentially expressed lincRNA (Figure 7A) were modulated at the MPP2-CLP transition, but not through the MPP3-CLP transition, suggesting that they might also be implicated in the differentiation of other hematopoietic lineages.

Even though the role of many of the identified lncRNAs is not yet known and requires further systematic investigations, most of them are involved in cancer progression or used as markers for cancer detection, emphasizing the importance of these molecules in the homeostatic balance. In particular, several members of the small nucleolar RNA host gene (SNHG) family are oncogenes [55] that play a crucial role in leukemogenesis and have prognostic value in acute myeloid leukemia [26]. In addition, high expression of SNHG7 promotes proliferation, metastasis, predicts poor prognosis and poor survival for patients.

Consistent with our results, Gm20645 and *Neat1* were found up- and downregulated, respectively, in the course of differentiation from HSC to MPP4 [37].

Furthermore, we identified 45 novel lncRNAs equally distributed between intergenic and intronic regions. The high expression of six of them in lymphoid progenitors as well as their predicted stable secondary structure, correlate with a potential function. Their identification constitutes a great breakthrough resulting from this work that would certainly require further deep investigation to measure specifically their biological role.

Altogether, our publicly available data represent a comprehensive resource for the lymphoid lineage commitment field that definitely will improve our understanding of the transcriptional organization underlying lineage specification.

## 4. Materials and Methods

### 4.1. Mice

C57Bl/6 Ly5.1 Thy1.2 (B6) mice were used at 10–18 weeks old (both males and females). They were purchased from CDTA (Orléans, France) and were kept in a specific pathogen-free facility of SFR Necker (Agreement No. 75-1026). All experimental procedures using animals were approved by the “Comité d’éthique en experimentation animale” of the Paris Descartes University and the French Ministry of Research, Innovation and Education under the following reference APAFIS #26599-2019071812345604.

### 4.2. Antibodies

For cell sorting, antibodies were obtained from BD Biosciences (San Diego, CA, USA), eBioscience (San Diego, CA, USA), or SONY (Weybridge, UK): anti-NK1.1 (PK136), anti-TCRb (H57-597), anti-CD3 (145-2-C11), anti-Mac 1/CD11b (M1/70), anti-CD19 (1D3), anti-GR 1 (RB6-8C5), anti-Ter 119 (Ly76), anti-Flt 3/CD135 (Flk2; A2F 10.1), anti-c-KIT/CD117 (SCF receptor; 2B8), anti-Sca 1 (E13-161.7), anti-IL7Ralpha/CD127 (A7R34), anti-VCAM1/CD106 (429), anti-CD138/Sdc1 (MI15), anti-CD93 (AA4.1). They were directly coupled to FITC, APC, APC-eF780, PE, PerCP-Cy5.5, BV421 or conjugated with biotin, the latter being revealed by streptavidin-PE-Cy7. All staining was performed for 20 min, at 4 °C with agitation.

### 4.3. Isolation of Bone Marrow Progenitors

BM cells from six mice were collected by crushing femurs, tibia, hips, and spine bones and pooled. The cells were incubated with unconjugated rat antibodies against B220, Ter119, Gr1, and Mac1. Positive cells were magnetically depleted with sheep anti-rat IgG-conjugated beads and sheep anti-mouse IgG-conjugated beads (Dynabeads M-450; Dynal Thermos Fisher Scientific, Villebon, France). The remaining cells (Lin^−^, enriched) were labeled with fluorescent conjugated antibodies against c-Kit, Sca-1, and Lin antigenes (NK1.1, TCRβ, CD3, Mac1, CD19, GR1, and Ter119). Lin^-^Sca-1^+^c-Kit^+^ cells are called LSK. Anti-VCAM1 and anti-Flt3 were additionally used to isolate the MPP2 (LSK-Flt3^+^VCAM1^+^) and the MPP3 (LSK-Flt3^+^VCAM1^−^) subsets. The CLP population is characterized by the addition of anti-IL7Ralpha and their phenotype is Lin^-^Sca-1^lo^c-Kit^lo^IL7Ralpha^+^ (Figure 1B). Cell sorting was performed on a FACS Aria III upgraded with DIVA software (BD Biosciences, San Diego, CA USA). Three independent biological replicates of MPP2, MPP3, and CLP were considered for transcriptome analysis. Pelleted cells were stored at −80 °C in QIAzol lysis reagent until RNA extraction.

### 4.4. Total RNA Isolation and Sequencing

Total RNA was extracted using miRNeasy Micro Kit (QIAGEN, Les Utils, France) with DNAse digestion. NanoDrop spectrophotometer was used to determine total RNA quantity and the 2100 Bioanalyser (Agilent) to qualify total RNA and subsequent libraries. Nuclear and mitochondrial ribosomal RNA was depleted using Ribo-Zero Gold rRNA Removal Kit (Epicentre). Whole transcriptome strand-specific libraries from three progenitors (MPP2, MPP3, and CLP) were performed using SMARTer Stranded RNAseq Kit (Clontech) with a total input of (0.11–0.33) µg of total RNA per cell population, knowing that rRNA can reach 90% of transcriptome according to cells. This library preparation allows the sequencing of RNAs coding for proteins and long non-coding RNAs (>~200 bp) polyA+ and polyA− (lncRNA, miRNA precursors, snRNA, snoRNA, etc.). Three biological replicates, per cell population, were sequenced on Illumina HiSeq2000 as 100 bp pair-end reads at high depth of coverage (2 × 120 million reads per replicate).

### 4.5. RT-PCR Analysis

Ten cells were collected in individual PCR tubes containing 5 µL PBS 1× following cell sorting on a FACSAria III U equipped with an automatic cell deposition unit (ACDU, BD Biosciences, San Diego, USA). Cells were lysed by freezing at −80 °C, followed by heating to 65 °C for 2 min.

After cooling at 4 °C, RNA was specifically reverse transcribed by adding 10 μL of a mix containing 0.13 μM RT primer, 1× PCR Buffer (Thermo), 3.3 mM MgCl_2_ (Thermo), 1 mM dNTPs (GE Healthcare), 40 units of RNaseOUT (Thermo), and 35 units of MuLV Reverse Transcriptase (Thermo), in a 15 μL reaction volume for 1 h at 37 °C and then incubated for 3 min at 95 °C for inactivation of reverse transcriptase. cDNA generated were dilute with 30 µL of 1× PCR Buffer.

A 4 µL aliquot of diluted cDNA was then amplified: first, a denaturation step at 95 °C for 10 min and then 54 amplification cycles (30 s at 95 °C, 45 s at 70 °C, and 60 s at 72 °C, with the hybridization temperature decreased from 70 to 60 °C during the six first cycles). The PCR mix contained 0.25 μM of each primer (Reverse and Forward), 1× PCR Buffer (Thermo), 2 mM MgCl_2_ (Thermo), 0.25 mM dNTPs (Thermo), and 0.5 units of AmpliTaq Gold DNA Polymerase (Thermo), in 20 μL reaction volume. PCR products were detected on a 1.5% agarose ethidium bromide gel. Primer sequences are described in Appendix A.

### 4.6. Culture of Lymphoid Precursors

After sorting, 1100 cells (CLP CD93^+^ or CLP CD93^−^) were seeded on a 24-well plate coated with 6000 OP9 or OP9-DL4 cells per well the day before. The co-culture was completed in Opti-MEM medium (Thermo) supplemented with 10% of FBS (Thermo), 100 U/mL of Penicillin-Streptomycin (Thermo), 50 µM β2mercaptoethanol, 1 ng/mL IL7 (Immunotools, Friesoythe, Germany), 5 ng/mL Flt3-L. Acquisition was performed 12 days later with a FACSCanto II (BD Biosciences). All cytometry analyses were performed with FlowJo™ Software V10.2 (BD Biosciences).

### 4.7. Comparison with Public Expression Data

Transcriptome data showing genes that increase in expression from MPP2 to CLP cells was intersected, with those of Cabezas et al. (2014) [37], whose expression increased in MPP4 cells compared to all the other populations (HSC, MPP1, MPP2, and MPP3) (Appendix A) and those of Klimmeck et al. (2014) [30] whose expression increased in LS + K cells compared to LS − K cells (Appendix A). These gene lists were explored by the PantherDB tool (Appendix A).

## Figures and Tables

**Figure 1 ijms-23-01115-f001:**
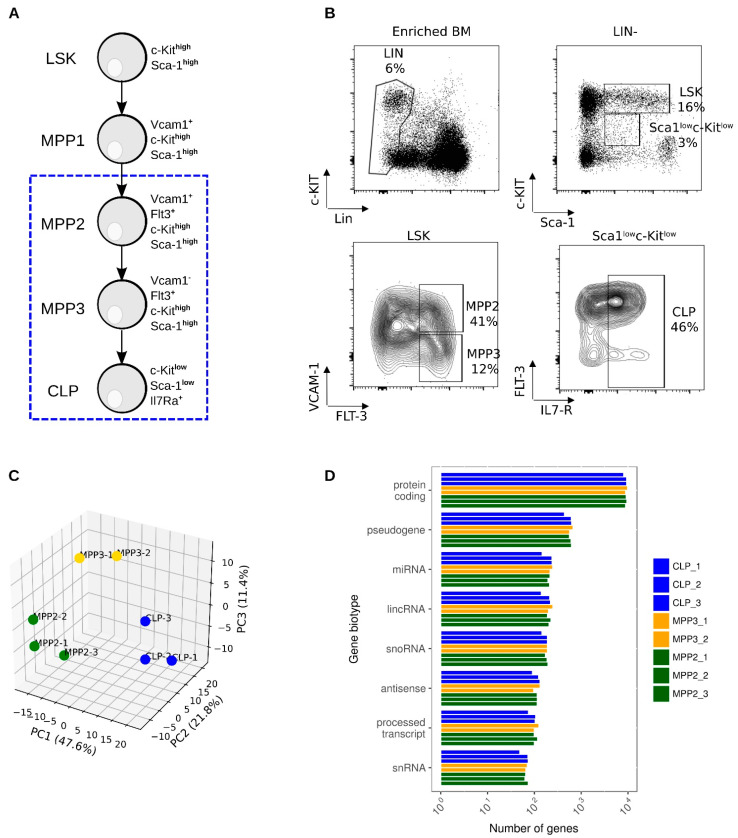
Isolation and total transcriptome sequencing of hematopoietic progenitor cells MPP2, MPP3, and CLP (**A**) Diagrammatic representation of the differentiation of LSK to MPPs and CLPs. The phenotype of each population is indicated. (**B**) FACS gates used to isolate BM progenitors: LSK (Lin^−^Sca1^+^c-Kit^+^), MPP2 (LSK, VCAM1^+^Flt3^+^), MPP3 (LSK, VCAM1^−^Flt3^+^), CLP (Lin-Sca1^low^c-Kit^low^IL7R^+^). Lin^−^ stands for lineage negative. One representative experiment out of nine is presented. The percentage (mean ± SEM) of each population obtained from nine experiments with about 20 mice each are: Lin^−^: 10.1 ± 1.6; LSK, within Lin^−^ cells: 10.6 ± 2.1; CLP: 41.9 ± 3.0 within Lin^−^c-Kit^low^Sca-1^low^. Within LSK: MPP1: 17.6 ± 1.8; MPP2: 34.6 ± 2.3; MPP3: 15.7 ± 2.0. (**C**) Principal component analysis (PCA) of gene expression values generated from biological replicates of MPP2, MPP3, and CLP. Expression data were normalized in FPKM (Fragments Per gene Kilobase and per Million reads), adjusted using ComBat algorithm before their clustering by PCA. (**D**) Bar plot showing the distribution of gene biotype quantification from gene expression values.

**Figure 2 ijms-23-01115-f002:**
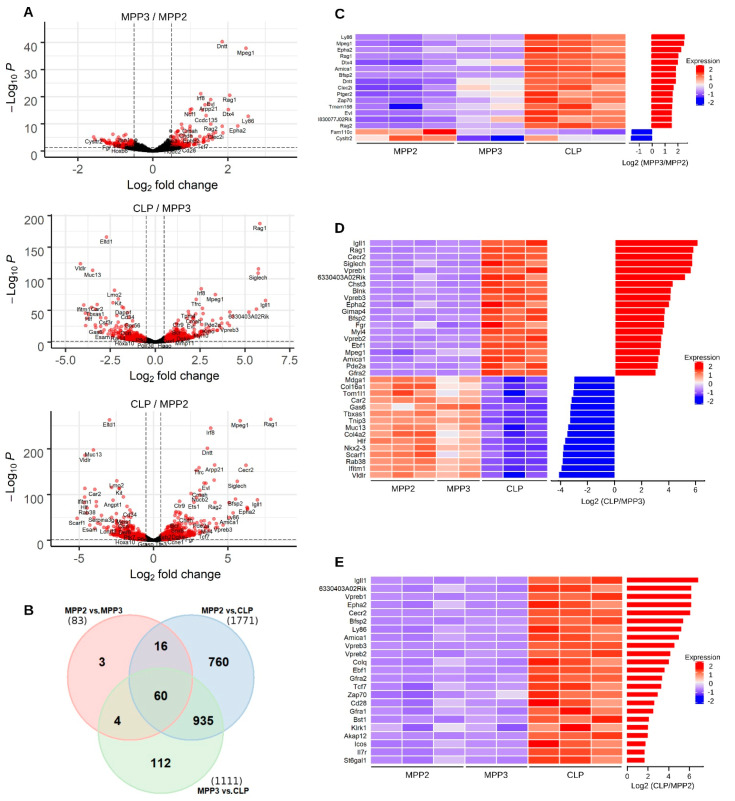
Differential gene expression analysis between progenitor cells. (**A**) Volcano plots showing for each gene (indicated by dots) the logarithm of the ratio expression levels between MPP2 and MPP3 (top panel), MPP3 and CLP (middle panel), MPP2 and CLP (bottom panel) according to the logarithm of the adjusted *p*-value generated from differential gene expression analysis. Red dots depict significantly under- or over-expressed genes for each comparison of progenitor cells. (**B**) Venn diagram illustrating the overlap of differentially expressed protein-coding genes identified from pairwise comparison between progenitor populations. (**C**,**D**) Heatmaps representing the abundance values of the most differentially expressed genes between MPP2 and MPP3, MPP3 and CLP cells, respectively. These genes were selected according to the logarithm of the ratio between their expression levels in the pairs of cell populations (fold change). The bar plots (right panel) illustrate the Log_2_ fold change per gene, for each pairwise comparison. (**E**) Heatmap showing the abundance values of the most differentially expressed genes between MPP2 and CLPs, associated with a very weak expression in MPP2. The bar plot (right panel) illustrates the Log_2_ fold change per gene.

**Figure 3 ijms-23-01115-f003:**
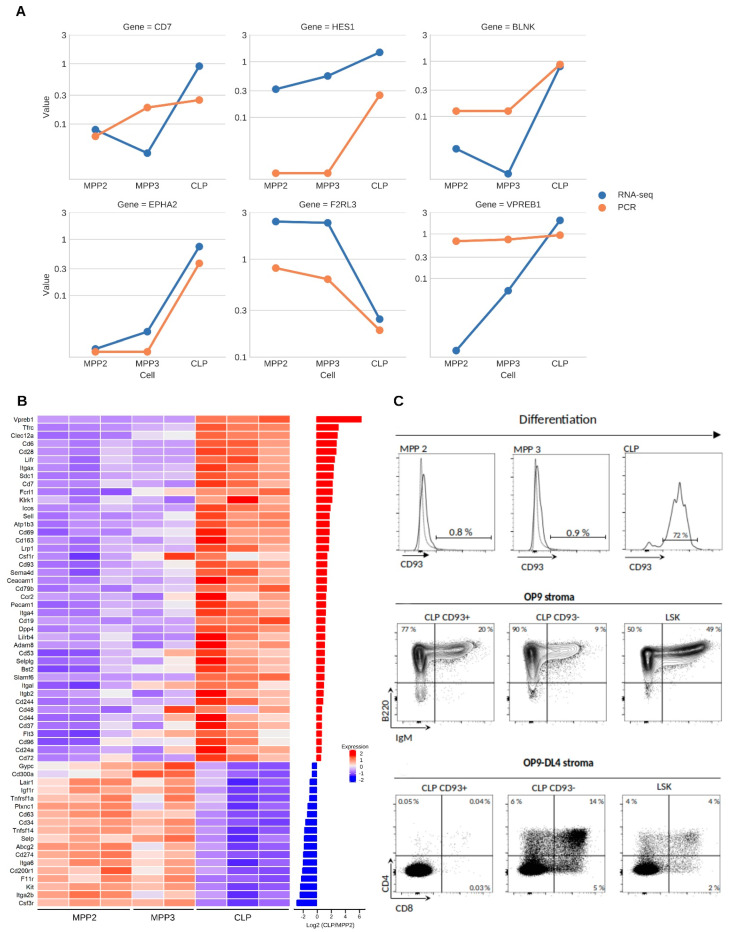
Exploration of surface markers and validation of CD93. (**A**) Line plots visualizing the temporal gene expression profiles generated from RNAseq (blue line) and RT-PCR (red line) experiments. The *y*-axis describes RNAseq data expressed as FPKM values, whereas RT-PCR corresponds to the percent of positive wells (0.1 corresponds to 10% whereas 1 corresponds to 100% of positive wells). (**B**) Heatmap showing the abundance values of the surface markers (CD) differentially expressed between MPP2 and CLP cells. The bar plot (right panel) illustrates the Log_2_ fold change per gene. (**C**) (Top panel) CD93 surface expression in BM progenitors (dotted lines represent the negative population). (Middle panel) In vitro differentiation of CD93^+^ and CD93^−^ CLP subsets compared with LSK. Sorted CD93^+^ and CD93 CLP were seeded on OP9 (B cell competent) stroma. (Bottom panel) In vitro differentiation of CD93^+^ and CD93^−^ CLP subsets compared with LSK. Sorted CD93^+^ and CD93^−^ CLP were seeded on OP9-DL4 (T cell competent) stroma. CD93^−^ CLP are able to generate B and T cells; in contrast, CD93^+^ CLP generate only B cells. Dot plots show the cells 21 days after co-culture.

**Figure 4 ijms-23-01115-f004:**
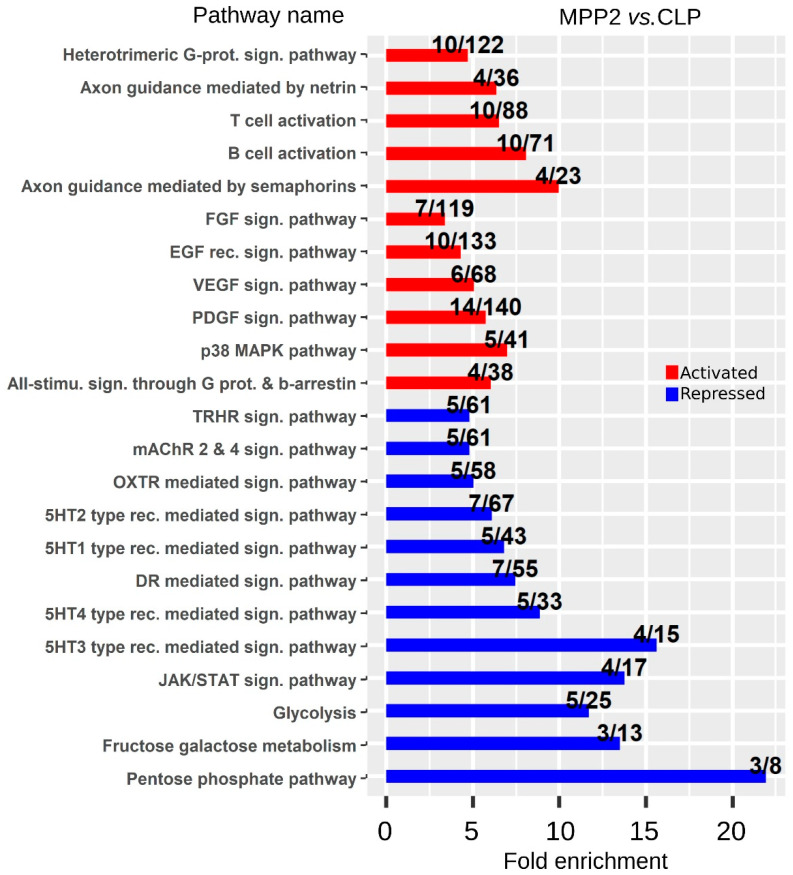
Biological pathways associated with lymphoid commitment. Bar plots illustrating pathway enrichment analysis from surface markers and transcription factors upregulated or downregulated between MPP2 and CLP cells. The pathways identified as specifically enriched from the group of genes upregulated between MPP2 and CLP were shown in red, those from downregulated genes were shown in blue. For each pathway, the bar plots illustrate the fold enrichment and the ratio describes the number of activated/repressed genes in the pathway compared to the total number of genes annotated in the pathway.

**Figure 5 ijms-23-01115-f005:**
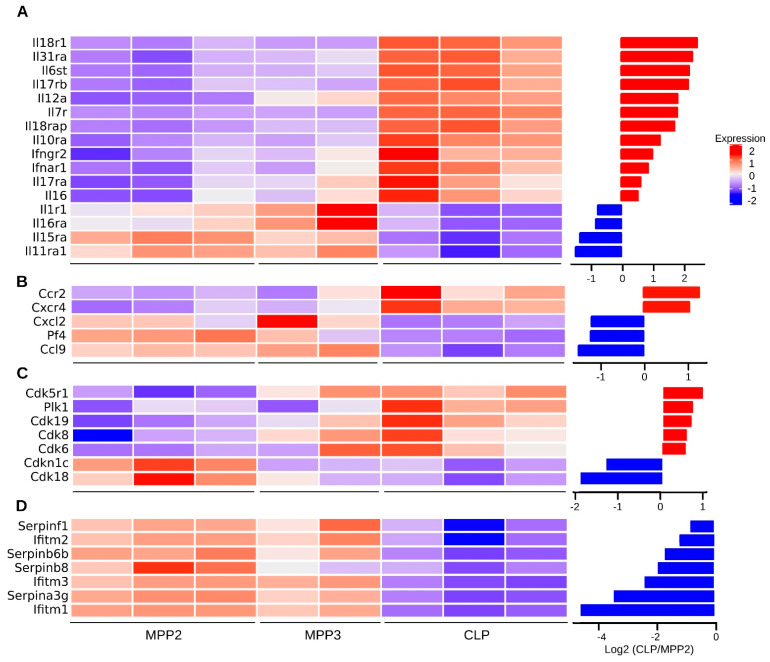
Expression analysis of functional group of genes. Heatmaps showing the abundance values of genes differentially expressed between MPP2 and CLP cells and associated with (**A**) interleukins and their receptors, (**B**) chemokines, (**C**) cell cycle proteins, (**D**) cellular integrity and stress. The bar plots (right panels) illustrate the Log_2_ fold change per gene.

**Figure 6 ijms-23-01115-f006:**
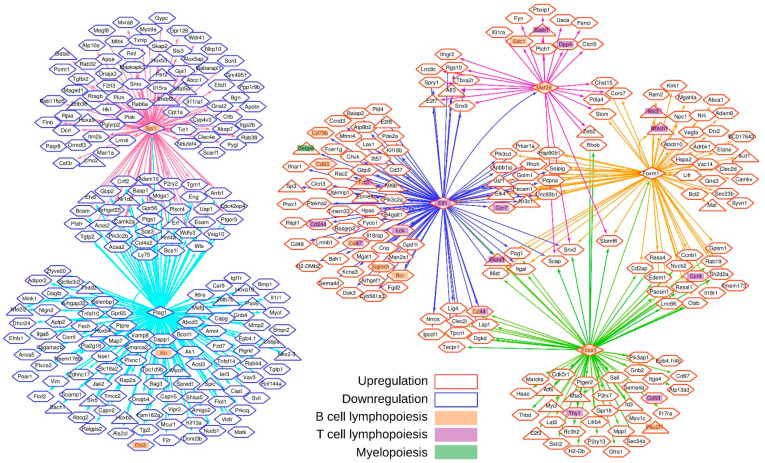
Gene regulatory network governing surface marker expression during lymphoid differentiation. Gene regulatory network inferred using gene co-expression, TF motif enrichment, and scanning algorithms. The node shapes represent transcription factors (triangles) or surface markers (hexagons). The node border colors annotate genes that are upregulated (red) or downregulated (blue) during lymphoid differentiation. The node background colors illustrate the genes known to be involved in B cell lymphopoiesis (orange), in T cell lymphopoiesis (purple), or in myelopoiesis (green). The edges indicate predicted regulatory interactions between core TFs and target genes. An arbitrary color was assigned to outgoing edges connected to the same TF in order to facilitate network visualization.

**Figure 7 ijms-23-01115-f007:**
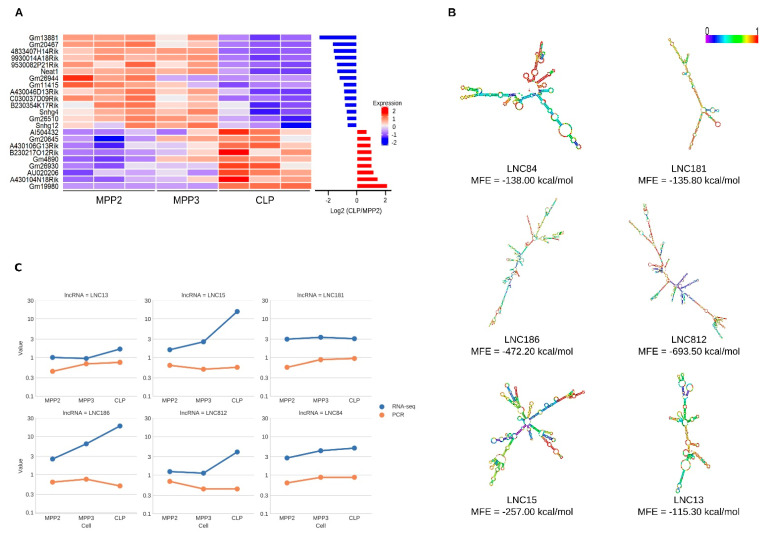
Annotated and novel long non-coding RNAs associated with lymphoid progenitor differentiation. (**A**) Heatmap showing the abundance values of the annotated long non-coding RNAs (lncRNA) differentially expressed between MPP2 and CLP cells. The bar plot (right panel) illustrates the Log_2_ fold change per gene. (**B**) Visualization of the predicted secondary structures of novel lncRNAs. The stability of lncRNA structures is calculated using MFE (minimum free energy). (**C**) Line plots visualizing the temporal gene expression profiles generated from RNAseq (blue line) and RT-PCR (red line) experiments. The *y*-axis describes RNAseq data expressed as FPKM values, whereas RT-PCR correspond to the percent of positive wells (0.1 corresponds to 10% whereas 1 corresponds to 100% of positive wells).

## Data Availability

The RNAseq data sets from MPP2-MPP3-CLP cells can be accessed through ArrayExpress (https://www.ebi.ac.uk/arrayexpress (accessed on 13 December 2021); accession number E-MTAB-11217).

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
