# Peer review of "Temporal Gene Expression Profiles Reflect the Dynamics of Lymphoid Differentiation"

_ijms, 2022, doi:10.3390/ijms23031115_

Round 1

Reviewer 1 Report

In this study, the authors reported the quantitative transcriptome of two heterogeneous multipotent progenitors’ subsets and the common lymphoid progenitor. Using deep RNA-sequencing of the whole transcriptome of three different lymphoid cell populations in mice, this study found potential new markers to improve purification of lymphoid populations. Some issues need to be addressed before this manuscript can be further evaluated. Here are my recommendations:

  1. Figure 1 to 7. The right parts of the figures were not completely shown. Please confirm the layout.
  2. Please check the citation style in the text.

Reviewer 2 Report

Chalabi et al present here results of a study on gene expression profile changes at specific stages of hematopoietic differentiation, focused on early lymphoid specification. By purifying 3 immunophenotypically different prgenitor cell subsets (corresponding to multipotent progenitors and common lymphoid progenitors) and studying the transcriptomes of each populations, the authors found difference in the expression of set of genes and genetic programs the mark the transition towards lymphoid commitment. The work is well designed and performed. Some of the finding from the transcriptome analysis have been confirmed by RTqPCR or by flow cytometry, increasing the value of the bioinformatic analysis. The authors also focused on lncRNAs at this stages of hematopoietic differentiation. Although this part of the work has not been further validated as the previous one, the novelty is worth reporting.

Round 2

Reviewer 1 Report

Thank you for the reply. All my concerns have been addressed.